# Message Passing Inference with Chemical Reaction Networks

**Nils Napp**
Wyss Institute for Biologically Inspired Engineering
Harvard University
Cambridge, MA 02138
nnapp@wyss.harvard.edu

**Ryan Prescott Adams**
School of Engineering and Applied Sciences
Harvard University
Cambridge, MA 02138
rpa@seas.harvard.edu

## Abstract

Recent work on molecular programming has explored new possibilities for computational abstractions with biomolecules, including logic gates, neural networks, and linear systems. In the future such abstractions might enable nanoscale devices that can sense and control the world at a molecular scale. Just as in macroscale robotics, it is critical that such devices can learn about their environment and reason under uncertainty. At this small scale, systems are typically modeled as chemical reaction networks. In this work, we develop a procedure that can take arbitrary probabilistic graphical models, represented as factor graphs over discrete random variables, and compile them into chemical reaction networks that implement inference. In particular, we show that marginalization based on sum-product message passing can be implemented in terms of reactions between chemical species whose concentrations represent probabilities. We show algebraically that the steady state concentration of these species correspond to the marginal distributions of the random variables in the graph and validate the results in simulations. As with standard sum-product inference, this procedure yields exact results for tree-structured graphs, and approximate solutions for loopy graphs.

## 1 Introduction

Recent advances in nanoscale devices and biomolecular synthesis have opened up new and exciting possibilities for constructing microscopic systems that can sense and autonomously manipulate the world. Necessary to such advances is the development of computational mechanisms and associated abstractions for algorithmic control of these nanorobots. Work on molecular programming has explored the power of chemical computation [3, 6, 11] and resulted in *in vitro* biomolecular implementations of various such abstractions, including logic gates [16], artificial neural networks [9, 10, 14], tiled self-assembly models [12, 15], and linear functions and systems [4, 13, 20]. Similarly, *in vivo* gene regulatory networks can be designed that when transformed into cells implement devices such as oscillators [8], intracellularly coupled oscillators [7], or distributed algorithms like pattern formation [1]. Many critical information processing tasks can be framed in terms of probabilistic inference, in which noisy or incomplete information is accumulated to produce statistical estimates of hidden structure. In fact, we believe that this particular computational abstraction is ideally suited to the noisy and often poorly characterized microscopic world. In this work, we develop a chemical reaction network for performing inference in probabilistic graphical models. We show that message passing schemes, such as belief propagation, map relatively straightforwardly onto sets of chemical reactions, which can be thought of as the "assembly language" of both *in vitro* and *in vivo* computation at the molecular scale. The long-term possibilities of such technology are myriad: adaptive tissue-sensitive drug delivery, *in situ* chemical sensing, and identification of disease states.

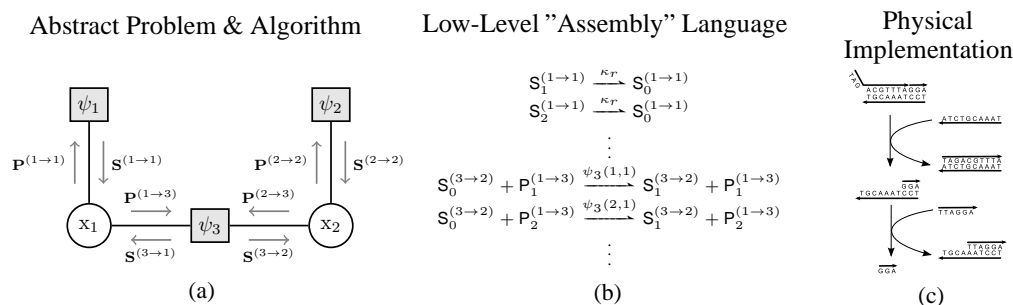

Figure 1: Inference at different levels of abstraction. (a) Factor graph over two random variables. Inference can be performed efficiently by passing messages (shown as gray arrows) between vertices, see Section 2. (b) Message passing implemented at a lower level of abstraction. Chemical species represent the different components of message vectors. The chemical reaction networks constructed in Section 3 perform the same computation as the sum-product message passing algorithm. (c) Schematic representation of DNA strand displacement. A given reaction network can be implemented in different physical systems, e.g. DNA strand displacement cascades [5, 17].

At the small scales of interest systems are typically modeled as deterministic chemical reaction networks or their stochastic counterparts that explicitly model fluctuations and noise. However, chemical reactions are not only models, but can be thought of as specifications or abstract computational frameworks themselves. For example, arbitrary reaction networks can be simulated by DNA strand displacement systems [5, 17], where some strands correspond to the chemical species in the specifying reaction network. Reactions rates in these systems can be tuned over many orders of magnitude by adjusting the toehold length of displacement steps, and high order reactions can be approximated by introducing auxiliary species. We take advantage of this abstraction by "compiling" the sum-product algorithm for discrete variable factor graphs into a chemical reaction network, where the concentrations of some species represent conditional and marginal distributions of variables in the graph. In some ways, this representation is very natural: while normalization is a constant concern in digital systems, our chemical design conserves species within some subsets and thus implicitly and continuously normalizes its estimates. The computation is complete when the reaction network reaches equilibrium. Variables in the graph can be conditioned upon by adjusting the reaction rates corresponding to unary potentials in the graph.

Section 2 provides a brief review of factor graphs and the sum-product algorithm. Section 3 introduces notation and concepts for chemical reaction networks. Section 4 shows how inference on factor graphs can be compiled into reaction networks, and in Section 5, we show several example networks and compare the results of molecular simulations to standard digital inference procedures.

To aid parsing the potentially tangled notation resulting from mixing probabilistic inference tools with chemical reaction models, this paper follows these general notational guidelines: capital letters denote constants, such as set sizes, and other quantities, such as tuples and message types; lower case letters denote parameters, such as reaction rates and indices; bold face letters denote vectors and subscripts elements of that vector; scripted upper letters indicate sets; random variables are always denoted by x or their vector version; and species names have a sans-serif font.

## 2 Graphical Models and Probabilistic Inference

Graphical models are popular tools for reasoning about complicated probability distributions. In most types of graphical models, vertices represent random variables and edges reflect dependence structure. Here, we focus on the *factor graph* formalism, in which there are two types of vertices that have a bipartite structure: variable nodes (typically drawn as circles), which represent random variables, and factor nodes (typically drawn as squares), which represent potentials (also called compatibility functions) coupling the random variables. Factor graphs, encode the factorization of a probability distribution and therefore its conditional independence structure. Other graphical models, such as bayesian networks, can be converted to factor graphs, and thus factor graph algorithms are directly applicable to other types of graphical models, see [2, Ch. 8].

Let $G$ be a factor graph over $N$ random variables $\{x_n\}_{n=1}^N$ where $x_n$ takes one of $K_n$ discrete values. The global $N$-dimensional random variable $\mathbf{x}$ takes on values in the (potentially huge) product space $\mathbf{K} = \prod_{n=1}^N \{1, ..., K_n\}$. The other nodes of $G$ are called factors and every edge in $G$ connects exactly one factor node and one variable node. In general, $G$ can have $J$ factors $\{\psi_j(\mathbf{x}^j)\}_{j=1}^J$ where we use $\mathbf{x}^j$ to indicate the subset of random variables that are neighbors of factor $j$, i.e. $\{x_n \mid n \in \text{ne}(j)\}$. Each $\mathbf{x}^j$ takes on values in the (potentially much smaller) space $\mathbf{K}^j = \prod_{n \in \text{ne}(j)} \{1, ..., K_n\}$, and each $\psi_j$ is a non-negative scalar function on $\mathbf{K}^j$. Together the structure of $G$ and the particular factors $\psi_j$ define a joint distribution on $\mathbf{x}$

$$\Pr(\mathbf{x}) = \Pr(x_1, x_2, \cdots, x_N) = \frac{1}{Z} \prod_{j=1}^J \psi_j(\mathbf{x}^j), \tag{1}$$

where $Z$ is the appropriate normalizing constant. Figure 1a shows a simple factor graph with two variable nodes and three factors. It implies that the the joint distribution $x_1$ and $x_2$ has the form $\Pr(x_1, x_2) = \frac{1}{Z}\psi_1(x_1)\psi_2(x_2)\psi_3(x_1, x_2)$.

The sum-product algorithm (belief propagation) is an dynamic programming technique for performing marginalization in a factor graph. That is, it computes sums of the form

$$\Pr(x_n) = \frac{1}{Z} \sum_{\mathbf{x} \backslash x_n} \prod_{j=1}^J \psi_j(\mathbf{x}^j). \tag{2}$$

For tree-structured factor graphs, the sum-product algorithm efficiently recovers the exact marginals. For more general graphs the sum-product algorithm often converges to useful approximations, in which case it is called *loopy belief propagation*.

The sum-product algorithm proceeds by passing "messages" along the graph edges. There are two kinds of messages messages from a factor node to a variable node and messages from a variable node to a factor node. In order to make clear what quantities are represented by chemical species concentrations in Section 4, we use somewhat unconventional notation. The $k$th entry of the *sum* message from factor node $j$ to variable node $n$ is denoted $S_k^{(j \to n)}$ and the entire $K_n$-dimensional vector is denoted by $\mathbf{S}^{(j \to n)}$. The $k$th entry of the *product* message from variable $n$ to factor node $j$ is denoted by $P_k^{(n \to j)}$ and the entire $K_j$-dimensional vector is denoted $\mathbf{P}^{(n \to j)}$. Figure 1a shows a simple factor graph with message names and their directions shown as gray arrows. Sum messages from $j$ are computed as the weighted sum of product messages over the domain $\mathbf{K}^j$ of $\psi_j$

$$S_k^{(j \to n)} = \sum_{\mathbf{k}_n^j = k} \psi_j(\mathbf{x}^j = \mathbf{k}^j) \prod_{n' \in \text{ne}(j)\backslash n} P_{\mathbf{k}_{n'}^j}^{(n' \to j)}, \tag{3}$$

where $\text{ne}(j)\backslash n$ refers to the variable node neighbors of $j$ except $n$ and $\mathbf{k}_n^j = k$ to the set of all $\mathbf{k}^j \in \mathbf{K}^j$ where the entry in the dimension of $n$ is fixed to $k$. Product messages are computed by taking the component-wise product of incoming sum messages

$$P_k^{(n \to j)} = \prod_{j' \in \text{ne}(n)\backslash j} S_k^{(j' \to n)}. \tag{4}$$

Up to normalization, the marginals can be computed from the product of incoming sum messages

$$\Pr(x_n = k) = \prod_{j \in \text{ne}(n)} S_k^{(j \to n)}. \tag{5}$$

The sum-product algorithm corresponds to fixed-point iterations that are minimizing the Bethe free energy. This observation leads to both partial-update or *damped* variants of sum-product, as well as asynchronous versions [18, Ch.6][19]. The validity of damped asynchronous sum-product is what enables us to frame the computation as a chemical reaction network. The continuous ODE description of species concentrations that represent messages can be thought of as an infinitesimally small version of damped asynchronous update rules.

## 3   Chemical Reaction Networks

The following model describes how a set of $M$ *chemical species* $\mathcal{Z} = \{Z_1, Z_2, \cdots, Z_M\}$ interact and their concentrations evolve over time. Each *reaction* has the general form

$$r_1\mathsf{Z}_1 + r_2\mathsf{Z}_2 + \cdots + r_M\mathsf{Z}_M \xrightarrow{\kappa} p_1\mathsf{Z}_1 + p_2\mathsf{Z}_2 + \cdots + p_M\mathsf{Z}_M. \tag{6}$$

In this generic representation most of the coefficients $r_m \in \mathbb{N}$ and $p_m \in \mathbb{N}$ are typically zero (where $\mathbb{N}$ indicates non-negative integers). The species on the left with non-zero coefficients are called *reactants* and are consumed during the reaction. The species on the right with non-zero entries are called *products* and are produced during the reaction. Species that participate in a reaction, i.e., $r_m > 0$, but where no net consumption or production occurs, i.e. $r_m = p_m$, are called *catalysts*. They change the dynamics of a particular reaction without being changed themselves.

A *reaction network* over a given set of species consists of a set of $Q$ reactions $\mathcal{R} = \{R_1, R_2, \cdots, R_Q\}$, where each reaction is a triple of reaction parameters (6),

$$R_q = (\mathbf{r}^q, \kappa_q, \mathbf{p}^q). \tag{7}$$

For example, in a reaction $R_q \in \mathcal{R}$ where species $\mathsf{Z}_1$ and $\mathsf{Z}_3$ form a new chemical species $\mathsf{Z}_2$ at a rate of $\kappa_q$, the reactant vector $\mathbf{r}^q$ is zero everywhere except at $\mathbf{r}_1^q = \mathbf{r}_3^q = 1$. The associated product vector $\mathbf{p}^q$ is zero everywhere except at $\mathbf{p}_2^q = 1$. In the reaction notation where non-participating species are dropped reaction $R_q$ is can be compactly written as

$$\mathsf{Z}_1 + \mathsf{Z}_3 \xrightarrow{\kappa_q} \mathsf{Z}_2. \tag{8}$$

### 3.1 Mass Action Kinetics

The concentration of each chemical species $\mathsf{Z}_m$ is denoted by $[\mathsf{Z}_m]$. A reaction network describes the evolution of species concentrations as a set of coupled non-linear differential equations. For each species concentration $[\mathsf{Z}_m]$ the rate of change is given by *mass action kinetics*,

$$\frac{d[\mathsf{Z}_m]}{dt} = \sum_{q=1}^{Q} \kappa_q \prod_{m'=1}^{M} [\mathsf{Z}_{m'}]^{\mathbf{r}_{m'}^q} (\mathbf{p}_m^q - \mathbf{r}_m^q). \tag{9}$$

Based on the fact that reactant coefficients appear as powers, the sum $\sum_{m=1}^{M} \mathbf{r}_m$ is called the *order* of a reaction. For example, if the only reaction in a network were the second order reaction (8), the concentration dynamics of $[\mathsf{Z}_1]$ would be

$$\frac{d[\mathsf{Z}_1]}{dt} = -\kappa_q[\mathsf{Z}_1][\mathsf{Z}_3]. \tag{10}$$

Similar to [4] we design reaction networks where the *equilibrium* concentration of some species corresponds to the results we are interested in computing. The reaction networks in the following section conserve mass and do not require flux in or out of the system, and therefore the solutions are guaranteed to be bounded. While we cannot rule out oscillations in general, the message passing methods these reactions are simulating correspond to an energy minimization problem. As such, we suspect that the particular reaction networks presented here always converge to their equilibrium.

## 4 Representing Graphical Models with Reaction Networks

In the following compilation procedure, each message and marginal probability is represented by a set of distinct chemical species. We design networks that cause them to interact in such a way that, at steady state, the concentration of some species represent the marginal distributions of the variable nodes in a factor graph. When information arrives the network equilibrates to the new, correct, value. Since messages in the sum-product inference algorithm are computed from other messages, the reaction networks that implement sending messages describe how species from one message catalyze the species of another message.

Beliefs and messages are represented as concentrations of chemical species: each component of a sum message, $\mathrm{S}_k^{(j \to n)}$, has an associated chemical species $\mathsf{S}_k^{(j \to n)}$; each component of a product message, $\mathrm{P}_k^{(n \to j)}$, has an associated chemical species $\mathsf{P}_k^{(n \to j)}$; and each component of a marginal probability distribution, $\Pr(x_n = k)$, has an associated chemical species $\mathsf{P}_k^n$. In addition, each message and marginal probability distribution has a chemical species with a zero subscript that represents unassigned probability mass. Together, the set of species associated with a messages or

marginal probability are called a *belief species*, and the reaction networks presented in the subsequent sections are designed to conserve species – and by extension their concentrations – with each such set. For example, the concentration of belief species $\mathcal{P}^n = \{P_k^n\}_{k=0}^{K_n}$ of $\Pr(x_n)$ have a constant sum, $\sum_{k=0}^{K_n}[P_k^n]$, determined by the initial concentrations. These sets belief species are a chemical representation of the left hand sides of Equations 3–5. The next few sections present reaction networks whose dynamics implement their right hand sides.

## 4.1 Belief Recycling Reactions

Each set of belief species has an associated set of reactions that recycle assigned probabilities to the unassigned species. By continuously and dynamically reallocating probability mass, the resulting reaction network can adapt to changing potential functions $\psi_j$, i.e. new information.

For example, the factor graph shown in Figure 1a has 8 distinct sets of belief species – 2 representing marginal probabilities of $x_1$ and $x_2$, and 6 (ignoring messages to leaf factor nodes) representing messages. The associate recycling reactions are

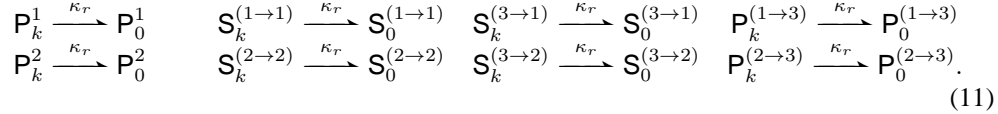

$$\tag{11}$$

By choosing a smaller rate $\kappa_r$ less of the probability mass will be unassigned at steady state, i.e. quantities will be closer to normalized, however the speed at which the reaction network reaches steady state decreases, see Section 5.

## 4.2 Sum Messages

In the reactions that implement messages from factor to variable nodes, message species of incoming messages catalyze the assignment of message species belonging to outgoing messages. The entries in factor tables determine the associated rate constants. The $k$th message component from a factor node $\psi_j$ to the variable node $x_n$ is implemented by a reactions of the form

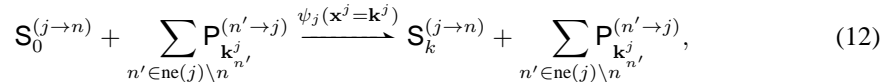

$$\tag{12}$$

where the $n$th component of $\mathbf{k}^j$ is clamped to $k$, $\mathbf{k}_n^j = k$. Using the law of mass action, the kinetics for each sum message species are given by

$$\frac{d[S_k^{(j\to n)}]}{dt} = \sum_{\mathbf{k}_n^j=k}\psi_j(\mathbf{x}^j = \mathbf{k}^j)[S_0^{(j\to n)}] \prod_{n'\in\text{ne}(j)\backslash n}[P_{\mathbf{k}_{n'}^j}^{(n'\to j)}] - \kappa_r[S_k^{(j\to n)}]. \tag{13}$$

At steady state the concentration of $S_k^{(j\to n)}$ is given by

$$\frac{\kappa_r}{[S_0^{(j\to n)}]}[S_k^{(j\to n)}] = \sum_{\mathbf{k}_n^j=k}\psi_j(\mathbf{x}^j = \mathbf{k}^j) \prod_{n'\in\text{ne}(j)\backslash n}[P_{\mathbf{k}_{n'}^j}^{(n'\to j)}], \tag{14}$$

where all $[S_k^{(j\to n)}]$ species concentrations have the same factor $\frac{\kappa_r}{[S_0^{(j\to n)}]}$. Their relative concentrations are exactly the message according to the to Equation (3). As $\kappa_r$ decreases, the concentration of unassigned probability mass decreases and the concentration normalized by the constant sum of all the related belief species can be interpreted as a probability. For example, the four factor-to-variable messages in Figure 1(a) can be implemented with the following reactions

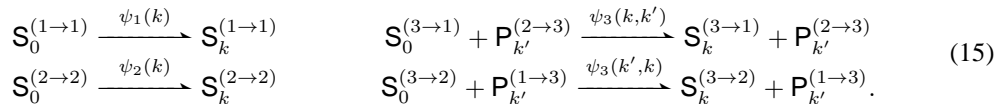

$$\tag{15}$$

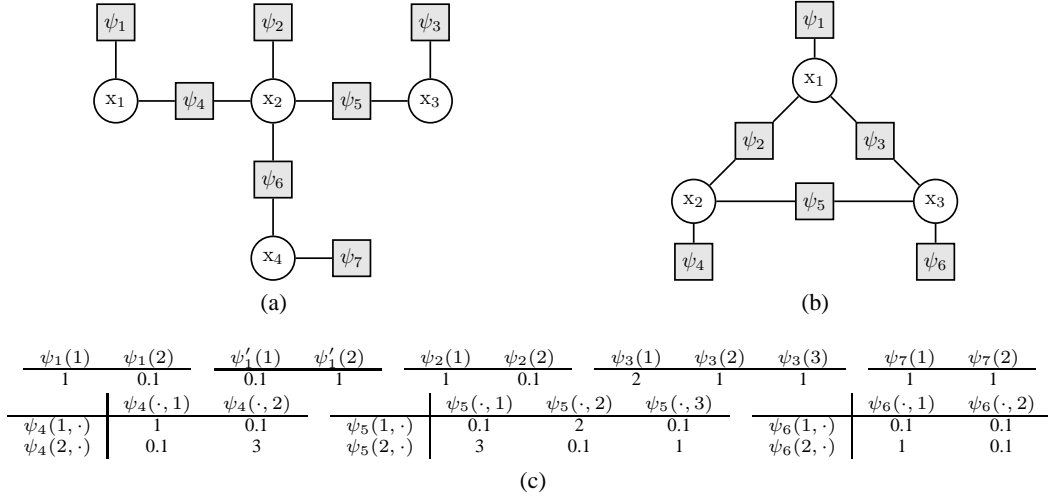

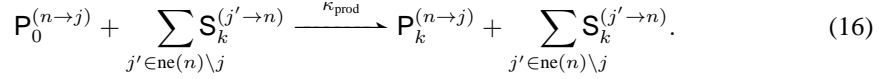

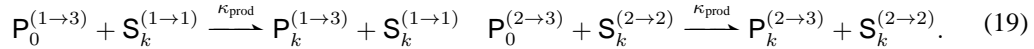

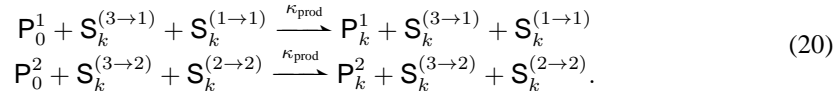

| $\psi_1(1)$ | $\psi_1(2)$ | $\psi_1'(1)$ | $\psi_1'(2)$ | $\psi_2(1)$ | $\psi_2(2)$ | $\psi_3(1)$ | $\psi_3(2)$ | $\psi_3(3)$ | $\psi_7(1)$ | $\psi_7(2)$ |
|---|---|---|---|---|---|---|---|---|---|---|
| 1 | 0.1 | 0.1 | 1 | 1 | 0.1 | 2 | 1 | 1 | 1 | 1 |

| | $\psi_4(\cdot,1)$ | $\psi_4(\cdot,2)$ | | $\psi_5(\cdot,1)$ | $\psi_5(\cdot,2)$ | $\psi_5(\cdot,3)$ | | $\psi_6(\cdot,1)$ | $\psi_6(\cdot,2)$ |
|---|---|---|---|---|---|---|---|---|---|
| $\psi_4(1,\cdot)$ | 1 | 0.1 | $\psi_5(1,\cdot)$ | 0.1 | 2 | 0.1 | $\psi_6(1,\cdot)$ | 0.1 | 0.1 |
| $\psi_4(2,\cdot)$ | 0.1 | 3 | $\psi_5(2,\cdot)$ | 3 | 0.1 | 1 | $\psi_6(2,\cdot)$ | 1 | 0.1 |

(c)

Figure 2: Examples of non-trivial factor graphs. (a) Four variable factor graph with binary factors. The factor leaves can be used to specify information about a particular variable. (b) Example of a small three variable cyclic graph. (c) Factors for (a) used in simulation experiments in Section. 5.1.

## 4.3 Product Messages

Reaction networks that implement variable to factor node messages have a similar, but slightly simpler, structure. Again, each components species of the message is catalyzed by all incoming messages species but only of the same component species. The rate constant for all product message reactions is the same $\kappa_{\text{prod}}$ resulting in reactions of the following form

$$\mathsf{P}_0^{(n \to j)} + \sum_{j' \in \text{ne}(n) \setminus j} \mathsf{S}_k^{(j' \to n)} \xrightarrow{\kappa_{\text{prod}}} \mathsf{P}_k^{(n \to j)} + \sum_{j' \in \text{ne}(n) \setminus j} \mathsf{S}_k^{(j' \to n)}. \tag{16}$$

The dynamics of the message component species is given by

$$\frac{d[\mathsf{P}_k^{(n \to j)}]}{dt} = \kappa_{\text{prod}}[\mathsf{P}_0^{(n \to j)}] \prod_{j' \in \text{ne}(n) \setminus j} [\mathsf{S}_k^{(j' \to n)}] - \kappa_r[\mathsf{P}_k^{(n \to j)}]. \tag{17}$$

At steady state the concentration of $\mathsf{P}_k^{(n \to j)}$ is given by

$$\frac{\kappa_r}{\kappa_{\text{prod}}[\mathsf{P}_0^{(n \to j)}]}[\mathsf{P}_k^{(n \to j)}] = \prod_{j' \in \text{ne}(n) \setminus j} [\mathsf{S}_k^{(j' \to n)}]. \tag{18}$$

Since all component species of product messages have the same multiplier $\frac{\kappa_r}{\kappa_{\text{prod}}[\mathsf{P}_0^{(n \to j)}]}[\mathsf{P}_k^{(n \to j)}]$, the steady state species concentrations compute the correct message according to Equation 4. For example, the two different sets of variable to factor messages in Figure 1a are

$$\mathsf{P}_0^{(1 \to 3)} + \mathsf{S}_k^{(1 \to 1)} \xrightarrow{\kappa_{\text{prod}}} \mathsf{P}_k^{(1 \to 3)} + \mathsf{S}_k^{(1 \to 1)} \quad \mathsf{P}_0^{(2 \to 3)} + \mathsf{S}_k^{(2 \to 2)} \xrightarrow{\kappa_{\text{prod}}} \mathsf{P}_k^{(2 \to 3)} + \mathsf{S}_k^{(2 \to 2)}. \tag{19}$$

Similarly, the reactions to compute the marginal probabilities of $x_1$ and $x_2$ in Figure 1a are

$$\begin{aligned}
\mathsf{P}_0^1 + \mathsf{S}_k^{(3 \to 1)} + \mathsf{S}_k^{(1 \to 1)} &\xrightarrow{\kappa_{\text{prod}}} \mathsf{P}_k^1 + \mathsf{S}_k^{(3 \to 1)} + \mathsf{S}_k^{(1 \to 1)} \\
\mathsf{P}_0^2 + \mathsf{S}_k^{(3 \to 2)} + \mathsf{S}_k^{(2 \to 2)} &\xrightarrow{\kappa_{\text{prod}}} \mathsf{P}_k^2 + \mathsf{S}_k^{(3 \to 2)} + \mathsf{S}_k^{(2 \to 2)}.
\end{aligned} \tag{20}$$

The two rate constants $\kappa_{\text{prod}}$ and $\kappa_r$ can be adjusted to tradeoff speed vs. accuracy, see Section 5.

Together, reactions for recycling probability mass, implementing sum-product messages, and implementing product messages define a reaction network whose equilibrium computes the messages and marginal probabilities via the sum-product algorithm. As probability mass is continuously recycled, messages computed on partial information will readjust and settle to the correct value. There is a clear dependence of messages. Sum messages from leaf nodes do not depend on any other messages. Once they are computed, i.e. the reactions have equilibrated, the message species continue to catalyze the next set of messages until they have reached the the correct value, etc.

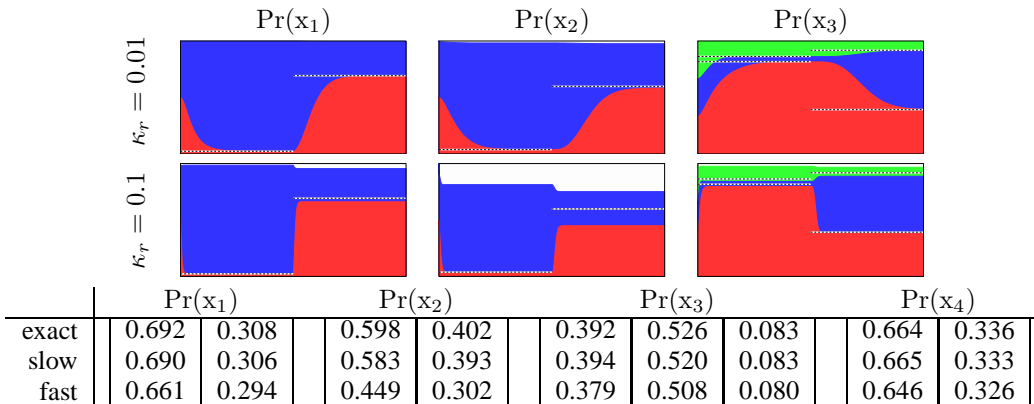

| | Pr(x₁) | | Pr(x₂) | | Pr(x₃) | | | Pr(x₄) | |
|---|---|---|---|---|---|---|---|---|---|
| exact | 0.692 | 0.308 | 0.598 | 0.402 | 0.392 | 0.526 | 0.083 | 0.664 | 0.336 |
| slow | 0.690 | 0.306 | 0.583 | 0.393 | 0.394 | 0.520 | 0.083 | 0.665 | 0.333 |
| fast | 0.661 | 0.294 | 0.449 | 0.302 | 0.379 | 0.508 | 0.080 | 0.646 | 0.326 |

Figure 3: Inference results for factor graph in Figure 2(a). Colored boxes show the trajectories of a belief species set in a simulated reaction network. The simulation time (3000 sec) is along the $x$–dimension. Half way though the simulation the factor attached to $x_1$ changes from $\psi_1$ to $\psi_1'$, and the exact marginal distribution for each period is shown as a back-white dashed line. The white area at the top indicates unassigned probability mass. These plots show the clear tradeoff between speed (higher value of $\kappa_r$) and accuracy (less unassigned probability mass). The exact numerical answers at 3000 sec are given in the table.

## 5 Simulation Experiments

This section presents simulation results of factor graphs that have been compiled into reaction networks via the procedure in Section 4. All simulations were performed using the SimBiology Toolbox in Matlab with the "sundials" solver. The conserved concentrations for all sets of belief species were set to 1, so plots of concentrations can be directly interpreted as probabilities. Figure 2 shows two graphical models for which we present detailed simulation results in the next two sections.

### 5.1 Tree-Structured Factor Graphs

To demonstrate the functionality and features of the compilation procedure described in Section 4, we compiled the 4 variable factor graph shown in Figure 2a into a reaction network. When $x_1$, $x_2$, $x_3$ and $x_4$ have discrete states $K_1 = K_2 = K_4 = 2$ and $K_3 = 3$, the resulting network has 64 chemical species and 105 reactions. The largest reaction is of 5th order to compute the marginal distribution of $x_2$. We instantiated the factors as shown in Figure 2c and initialized all message and marginal species to be uniform. To show that the network continuously performs inference and can adapt to new information, we changed the factor $\psi_1$ to $\psi_1'$ half way through the simulation. In terms of information, the new factor implies that $\Pr(x_1 = 2)$ is suddenly more likely. In terms of reactions the change means that $\mathsf{S}_0^{(1\to1)}$ is now more likely to turn into $\mathsf{S}_2^{(1\to1)}$. In a biological reaction network, such a change could be induced by up-regulating, or activating a catalyst due to a new chemical signal. This new information changes the probability distribution of all variables in the graph and the network equilibrates to these new values, see Figure 3.

The only two free parameters are $\kappa_{\mathrm{prod}}$ and $\kappa_r$. Since only $\kappa_r$ has an direct effect on all sets of belief species, we fixed $\kappa_{\mathrm{prod}} = 50$ and varied $\kappa_r$. Small values of $\kappa_r$ results in better approximation as less of the probability mass in each belief species set is in an unassigned state. However, small values of $\kappa_r$ slow the dynamics of the network. Larger values of $\kappa_r$ result in faster dynamics, but more of the probability mass remains unassigned, top white area in Figure 3. We should note, that at equilibrium, the relative assignments of probabilities are still correct, see Equation 14 and Equation 18.

The compilation procedure also works for factor graphs with larger factors. When replacing the two of the binary factors $\psi_5$ and $\psi_6$ in Figure 2a with a new tertiary factor $\psi_5'$ that is connected to $x_2$,$x_3$, and $x_4$ the compiled reaction network has 58 species and 115 reactions. The largest reaction is of order 4. Larger factors can reduce the number of species since there are fewer edges and associated messages to represent, however, the domain sizes $\mathbf{K}^j$ of the individual factors grows exponentially and in the number of neighbors and thus require more reactions to implement.

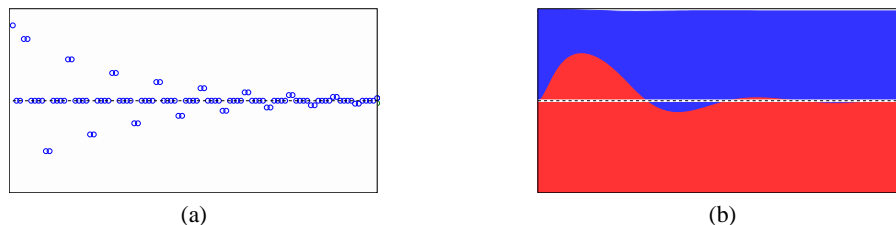

<div align="center">(a)                                  (b)</div>

Figure 4: (a) The belief of $\Pr(x_1 = 1)$ as function of iteration in loopy belief propagation. All messages are updated simultaneously at every time step. After 100 iterations the oscillations abate and the belief converges to the correct estimate indicated by the dashed line. (b) Trajectory of $\mathsf{PA}_i$ species concentrations. The simulation time is 3000 sec and the different colors indicate the belief of about either of the two states. The dotted line indicates the exact marginal distribution of $x_1$.

## 5.2 Loopy Belief Propagation

These networks can also be used on factor graphs that are not trees. Figure 2b shows a cyclic graph which we compiled to reactions and simulated. When $K_n = 2$ for all variables the resulting reaction network has 54 species and 84 reactions. We chose factor tables that anti-correlate neighbors and leaf factors that prefer the same state.

Figure 4 shows the results of performing both loopy belief propagation and simulation results for the compiled reaction network. Both exhibit decaying oscillations, but settle to the correct marginal distribution. Since the reaction network is essentially performing damped loopy belief propagation with an infinitesimal time step, the reaction network implementation should always converge.

## 6 Conclusion

We present a compilation procedure for taking arbitrary factor graphs over discrete random variables and construct a reaction network that performs the sum-product message passing algorithm for computing marginal distributions.

These reaction networks exploit the fact that the message structure of the sum-product algorithm maps neatly onto the model of mass action kinetics. By construction, conserved sets of belief species in the network perform implicit and continuous normalization of all messages and marginal distributions. The correct behavior of the network implementation relies on higher order reactions to implement multiplicative operations. However, physically high order reaction are exceedingly unlikely to proceed in a single step. While we can simulate and validate our implementation with respect to the mass action model, a physical implementation will require an additional translation step, e.g. along the lines of [17] with intermediate species of binary reactions.

One aspect that this paper did not address, but we believe is important, is how parameter uncertainty and noise affect the reaction network implementations of inference algorithms. Ideally, they would be robust to parameter uncertainty and random fluctuations. To address the former one could directly compute the parameter sensitivity in this deterministic model. To address the latter, we plan to look at other semantic interpretations of chemical reaction networks, such as the linear noise approximation or the stochastic chemical kinetics model.

In addition to further analyzing this particular algorithm we would like to implement others, e.g. max-product, parameter learning, and dynamic state estimation, as reaction networks. We believe that statistical inference provides the right tools for tackling noise and uncertainty at a microscopic level, and that reaction networks are the right language for specifying systems at that scale.

### Acknowledgements

We are grateful to Wyss Institute for Biologically Inspired Engineering at Harvard, especially Prof. Radhika Nagpal, for supporting this research. We would also like to thank our colleagues and reviewers for their helpful feedback.

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
