[Reviews · NeurIPS 2013]

Submitted by Assigned_Reviewer_4

Summary:
The authors present a compilation procedure for implementing message passing for factor graphs over discrete random variables in a chemical reaction network. A chemical reaction network consists of a set of reactions, comprised of reactants, products and reaction rates.
They use mass action kinetics to design and implement specific information flows through chemical species.
The authors show how the sum-product algorithm can be mapped onto a set of reactions for computation of sum messages, product messages and belief recycling reactions. Furthermore, it is discussed how global equilibrium states can be achieved and how these correspond to the states BP would achieve in a graphical model.


The paper is very well written, with very minor occasional typos or missing prepositions. However, the notation for the chemical reaction networks is unusual to the machine learning community and takes significant effort to read comfortably. As this is a byproduyct of the unusual topic, it is not a mistake of the authors, but needs to be kept in mind if targeting ML communities. Still, the authors try to find mathematical abstractions for chemical reactions and explain them cleanly.


Content-wise, this is an unusual paper for the machine learning community. the mapping of a fundamental inference algorithm to chemical reaction networks is presented in a lot of detail, but can be confusing in terms of certain aspects: it is a bit unclear how exactly damped BP corresponds to what the authors are doing as they are stating and I would have appreciated a little more elaboration on it. Furthermore, the mentioned problem of unassigned probability mass in case of high reaction speeds is a recurrent topic in the paper and could use a dedicated paragraph where it is theoretically explained in one concentrated place to reduce confusion about it. The authors make the strong statement that arbitrary factor graphs can be implemented, but only show experiments for very low dimensional variables and comparably small graphs. A larger scale implementation of a complicated model and a more detailed quantitative comparison compared to the shown experiments would be welcome. The proposed experiments underline the suggested contribution, but are not very extensive or deep. Considering this is work at an early stage, it is reasonable that the authors suggest exploring noisy and uncertain settings. It would be exciting to see a non-toy experiment (i.e. labeling in an mrf) implemented through chemical reaction networks and compare it to real BP solutions, probably this should happen with the introduction of the Max-Product algorithm the authors suggest for the future.
In total, the paper is meticulous in suggesting the framework of chemical reaction networks and mapping belief propagation to it, but the experiments appear a little lacking in real scope.
Suggestions:
a)more and more convincing experiments with more complicated and/or bigger graphs
b)better theoretical explanation of damped BP in relation to this work
c) discuss how reaction speeds can be implemented in reality with different kappas.
I expect them to be regulated through chemical compounds, which would most likely lead to discrete subsampling of the speed-space. Would this lead to local minima or other problems during inference? Are the assumptions of the 'perfect chemical reaction network' based on arbitrary species realistic? Where's the catch if graphs get bigger and have largewr state spaces and hundreds/thousands of chemical species are needed to implement a problem. Does it scale?
d) Discussions of more models. MaxProduct, continuous variables etc....



I found the paper to be highly original for the ML-community. While biological computation is a vibrant field with many recent contributions by Erik Winfree, Luca Cardelli, Andrew Phillips and more researchers stuydying multiple aspects of DNA computing and Biological Programming, the particular focus of this work in implementing a standard ML inference algorithm and mapping it cleanly to a chemical implementation is new to the best of my knowledge and highly significant. It will be of great value to the field of biological computation and computation in general if well-established and theoretically studied inference algorithms can be mapped to biological/hardware implementations, especially considering the possibilities of microscopic computation in the future.
Summary: In conclusion, I find this to be an intriguing paper and encourage the authors to build on this work. While it does not provide novel theoretical insights into probabilistic machine learning and is not complete in its experimental evidence or the amount of algorithms that have been explored, it offers a formal mapping of belief propagation to biological computation and is such a solid early step into an exciting field that machine learning could explore in the future.

Update: After the rebuttals I have upgraded my score to reflect the authors responses to my concerns.


Submitted by Assigned_Reviewer_5

This paper shows that the loopy propagation can be implemented as a chemical reaction network. Although the topic of this paper is completely novel, it is a little difficult to evaluate this paper, because of the following two reasons.

1) The experiment is only done by computational simulations.
2) The purpose of implementing chemical reaction networks is not clear.

If it is really implemented in DNAs, it would be much more impressive (and probably should be submitted to life science journals). I think the computational part itself is not so impressive, but if it is really implemented, it can be a great achievement.

The fact that the chemical reaction network can be seen as a loopy propagation is interesting. It would be great if an existing metabolic network can be interpreted as message passing, because it opens a way to engineer the network, e.g., to produce useful substances.
Summary: The idea is novel, but the drawback is that it is not implemented in vitro.

Submitted by Assigned_Reviewer_6

New comment: I acknowledge that I have read authors' rebuttals and other reviewers' comments. The rebuttals are precise and valid. I still hold my judgment that this paper is very novel and interesting in both machine learning and synthetic biology communities. I suggest NIPS to accept this paper.

This paper proposed a procedure to compile an arbitrary probabilistic graphical model in the form of a factor graph over discrete random variables into a chemical reaction network. They further showed that the steady state concentrations of the species in the network are the same to the marginal distributions of random variables in the graph. This is a very interesting and well written paper. I have the following comments:
1. Although the idea is novel and interesting, it compiles a probabilistic graphical model into a chemical reaction networks with much more species than the random variables. However, simulations and implementations of chemical reaction networks are known to be difficult and time-consuming. Therefore, what are the possible applications of this procedure? I can see this procedure nicely connects two important concepts in machine learning and synthetic biology, but how can this be applied to advance or solve the key problems in either community?
2. The current procedure cannot deal with continuous random variables. Could the authors discuss the possible direction for continuous random variables, which are commonly used in probabilistic graphical models?
3. The reactions are assumed to be mass action kinetics in the paper. If other reaction types are used, such as Michaelis-Menten kinetics, can the procedure still work?
4. In Eq. 3, k^j_n is not defined.
5. There is typo in Eq. 11.
6. Why did you ignore messages to leaf factor nodes in the procedure?
7. In Fig 3 caption, grey color is mentioned, but in the figure, there is no grey color.
Summary: This is a very interesting paper that nicely connects two key concepts in machine learning and synthetic biology together.
Author Feedback

Author rebuttal: First, we appreciate the encouraging feedback and would like to thank the reviewers for their helpful comments. We believe that bringing inference algorithms to the scale of molecular machines, which are notoriously plagued by noise and weak information, is an exciting opportunity for expanding the scope of machine learning techniques. We intend this paper to be a first step in that direction and thus focused on an algorithm that is basic to ML and map it onto computations that are natural on the molecular scale. In contrast to much related work on molecular programming, we avoid intermediate computational abstractions, such as logic gates, artificial neurons, amplifiers, etc., and thus produce a novel, and rather direct, connection between two models that are fundamental to their respective fields: BP in machine learning and mass action kinetics in molecular programming.

Applications for a principled approach to managing uncertainty on a molecular scale are myriad, even for small factor graphs, for example: sensor fusion for noisy, leaky, or otherwise unreliable molecular receptors; engineering more robust cellular signaling; or estimating environmental conditions that are not directly observable by a molecular machine. All reviews mention the lack of substantial examples, and we agree that they would make the paper stronger. However, since we consider the direct translation of BP into the dynamics that govern molecular systems to be the primary contribution, we focused our efforts and limited space to make that connection as clear as possible, instead of producing extensive examples. That said an application specific example graph, such as a molecular sensor fusion problem, is an excellent suggestion for an improved revision.

While implementation is (currently) not our focus, we would like to point out that we use standard simulation tools and that, for in-vitro systems, the size is of our proposed reaction networks roughly in line with current state-of-the-art DNA implementations (72 distinct species, Qian et al, 368-72 NATURE Vol. 457, 2011). Regarding the issue of scaling, it is true that this direct approach will probably not scale well to large graphs. However, besides the potential utility of small networks a better “compiler” that takes into account expected operating regimes could eliminate many unimportant species (as we did with single factor messages to leaf nodes in the current draft). The work we present here produces a correct, but un-optimized network and various approximations might efficiently implement it.

Bridging two fields with different common notation, we obviously struggled with creating a clear presentation and appreciate both the readers’ patience their suggestions for improving readability. A dedicated paragraph on “unassigned” probability mass will help with the overall flow. Similarly, clarifying the connection between damped loopy BP and our proposed solution will make future versions of this paper more readable. In short, the steady state solution corresponds to one update step in BP. However, since the dynamics of mass action kinetics slowly and continuously approach this solution, one could interpret this behavior as an infinitesimal version of damped loopy BP.

We especially thank the reviewers for pointing out interesting extensions and would like to briefly comment on some of the ideas. We are very much interested in exploiting other reaction models (such as Michaelis-Menten) to implement computations in ML algorithms and suspect that there are applications, especially regarding the common use of exponentiation/log in ML algorithms which we found difficult to express in the mass action model. Inference with continuous RVs would be extremely useful for estimating concentrations! We plan to pursue this idea, but representing beliefs with molecular species and consequently the associated reaction networks would look rather different. Closely related algorithms like MaxProduct are much simpler extensions. Perhaps more farfetched but correspondingly impactful is the idea of interpreting biological reaction networks as inference machines. It is likely that microscopic organisms do perform some kind of inference, and that evolutionary tuning has created (by some measure) efficient implementations. Direct translations between ML algorithms and reaction networks, such as the one we present here, are a key step to enable such an analysis. Instead of simply providing tools for molecular engineers, understanding how bags of wiggling molecules perform inference represent an opportunity for basic advances in ML. Again, we view this paper as a first, but solid, step into a new area and feel that the many interesting extensions and follow-up questions support this point of view.